# Evaluating the Cooling Efficiency of Polymer Injection Molds by Computer Simulation Using Conformal Channels

**DOI:** 10.3390/polym15204044

**Published:** 2023-10-10

**Authors:** Carlos Vargas-Isaza, Adrian Benitez-Lozano, Johnnatan Rodriguez

**Affiliations:** 1Grupo Investigación Materiales Avanzados y Energía, Instituto Tecnológico Metropolitano, Medellín 050034, Colombia; carlosvargas@itm.edu.co; 2Grupo Investigación Calidad Metrología y Producción, Instituto Tecnológico Metropolitano, Medellín 050034, Colombia; adrianbenitez@itm.edu.co; 3Departamento de Ingeniería Mecánica, Universidad EIA, Envigado 055428, Colombia

**Keywords:** simulation, conformal cooling channels, additive manufacturing, steel, polycarbonate, cooling efficiency, warpage

## Abstract

Injection molds are production tools that require detailed analysis based on the quality of the resulting part, the impact on cycle times, and the expected production volume. Cooling channels also play a critical role in mold performance and product quality as they largely determine cycle time. Designs that incorporate conformal cooling channel (CCC) geometries that conform to or align with the part contour are currently being explored as an alternative to conventional cooling channel designs in injection molds. In this study, a simulation of CCC geometries was performed and their effects on mold temperatures and warpage were investigated. Two cross-sectional geometries, circular and square, were selected for a three-factor level design of experiments (DOE) analysis. The response variables used were mold temperatures and part warpage. A cup-shaped part with upper and lower diameters of 54 and 48 mm, respectively, a height of 23 mm and a thickness of 3 mm was used for the injection molded part. A comparison was also made between two materials for the injection mold, steel and polycarbonate. The DOE results showed that the distance between the CCC and the injected part and the diameter or side of the square have significant effects on the response variables for both systems (steel and polycarbonate molds). In addition, a comparison between conventional and conformal cooling channels was analyzed using a cup-shaped part and a less rigid part geometry. The finite element simulation results show a 9.26% reduction in final warpage in the cup-shaped part using CCCs compared with the conventional cooling methods in steel. When using parts with lower geometry stiffness, the use of CCCs reduced final part warpage by 32.4% in metal molds and by 59.8% in polymer molds.

## 1. Introduction

Injection molding is one of the most widely used techniques in plastics processing today. The quality of the molds is of paramount importance as it directly influences the geometry of the manufactured parts [1]. An important feature of injection molds is the cooling channels; traditionally, the cooling channels are made as a straight hole to cool the injected part. However, sometimes the cooling performance is not optimal because the distance between the cooling channel and the surface varies, resulting in different cooling rates and uneven temperature distribution [2]. With the development of additive manufacturing (AM), there is the possibility to build cooling channels that follow the contour of the injected parts in the mold; thus, they are named conformal cooling channels (CCCs) [3].

Computerized simulation has proven to be an indispensable tool for evaluating CCCs [3,4,5,6,7,8]. This tool enables mold designers to assess and demonstrate the advantages of adopting this injection mold design type over traditional molds with conventional cooling channels. Within the field of injection mold simulation, there is an abundance of literature and research focused on the integration of conformal channels. This effort leverages a range of commercial software solutions, including but not limited to Moldex3D^®^, Moldflow^®^, and Solidworks^®^ Plastic [4,5,6]. These platforms are purpose-built to simulate the complete injection process holistically, including its myriad variables. Moreover, they facilitate fluid dynamic computational analyses, akin to those conducted using ANSYS, all rooted in firmly established injection process boundary conditions [7,8]. The advantages of CCCs compared with conventional channels in injection molds are evident through both simulation tools and experimental assemblies. These improvements are seen in terms of achieving a more uniform cooling of the mold, resulting in reduced cooling times, shrinkages, warpages, and residual stresses in the injected part [9,10,11,12].

Figure 1a,b summarize two important criteria in the simulation and evaluation of CCCs, the injection molding materials used and the input variables or design parameters of the CCCs, respectively. This report was compiled from a review of several research papers discovered through a bibliographic search of the Scopus database on computer simulation of the injection process using CCCs. Figure 1a illustrates that the majority of simulation studies focus on steel molds and their various alloys [13,14,15], due to their high conductivity and durability. Mold materials, such as epoxies [16,17,18], reinforced liquid silicones [19,20], and other types of polymers [21,22], have begun to be used as CCC substitutes in well-known rapid tooling applications for low-volume injection molding. However, alternative mold materials such as polycarbonate have not been used to date. 

Several analyses have been performed based on the input variables or design parameters of the CCCs. First, specific CCC configurations have been evaluated under different injection process conditions [8,15,23,24,25]. Furthermore, different CCC configurations have been evaluated considering different values of channel diameters, channel spacings (P) and channel distances to the cavity cooling wall (L) [4,7,9,13,14,26]. Many of these analyses focused on parts with high geometric complexity or features that make conventional channel cooling inefficient. This includes parts with considerable depth and curved contours, as well as those whose dimensions or shape make them more susceptible to deformation during injection and demolding [4,5,6,7,9]. The current study evaluates all the parameters usually examined in this type of analysis, unlike prior studies which focused on only a few parameters. These include cross-sectional shape, channel diameters, step (P), wall distance (L), and the layout of the CCCs. Our study expands on the analysis by examining the effects of substantial differences in mold materials (thermal conductivity) and the geometries of injected parts (refer to the parameters marked with an asterisk in Figure 1b).

The manufacture of products with complex geometries and the addition of material layer by layer is referred to as additive manufacturing (AM) [27,28]. A variety of materials are currently being used in 3D printing, including metals, polymers, ceramics, and composites [27,29,30,31]. Recently, AM technologies have been used to create complex but efficient cooling channels for injection molding. Conventional cooling channels are machined in straight lines and are usually circular in cross-section. However, straight channels do not provide reliable cooling across the mold surface. Uneven cooling leads to longer cycle times and warpage [32]. Cooling performance can be improved by mold material properties and the efficiency of the cooling system. The choice of material for the mold is based on mechanical properties such as strength and weight, among others. Therefore, cooling channels can follow the contours of the mold cavity to improve cooling efficiency. When compared with conventional channels, CCC can achieve a 15–50% reduction in cooling time [3,33]. Mazur et al. [34] used selective laser melting to fabricate CCCs, addressing the manufacturability of the channels with circular and self-supporting cross-sections and evaluating the roughness and stress concentration. It was found that the dimensional accuracy of the circular sections was compromised with variation in inclinations. The top surface of the channel had greater roughness, and a plot of stress concentration versus cooling variation was presented as a design tool. Kuo et al. [33] achieved an 81% reduction in cooling time by manufacturing CCCs with a semicircular cross-section in a spiral configuration in the mold cavity and longitudinal channels with a rectangular cross-section in the mold core, demonstrating the feasibility of using CCC features in both parts of the mold.

The efficiency and cycle time of an injection molding process are affected by several factors, including mold geometry, material selection, cooling channel position, and flow rate. By optimizing these factors, the efficiency of the process can be improved and cycle times can be reduced, resulting in high quality end products and increased profitability [35]. Jahan S and -El-Mounayri [36] used a numerical model to predict the cycle time. Then, the numerical model was validated experimentally, and the effect of several design parameters on the CCC performance was studied using the design of experiments (DOE) approach. It was found that the best performance was achieved with a 6 mm diameter conformal cooling channel with a pitch of 8 mm and a channel centerline to mold wall distance of 4 mm. In addition, considering the thermal and mechanical performance of the mold with CCCs, the configuration for optimal design was 6 mm diameter, 12 mm pitch, and 6 mm channel centerline to mold wall distance [36]. Melt temperature, injection time, packing pressure, packing time, cooling time, and cooling temperature were used as design variables to study cooling performance, cycle time, and warpage in a multi-objective optimization of process parameters [37]. Based on the numerical and experimental results, the CCCs were found to be effective in reducing cycle time and warpage. Venkatesh G and Ravi Kumar Y. [37] used a Taguchi method and finite element analysis to obtain the optimum design of CCCs for thermal stress in the mold. The channel cross-section (circle, trapezoid, rectangle), cooling path (spiral, spiral-square, straight) and center line (19, 25, 31 mm) were used as input variables for the statistical model. The results showed that the most influencing parameter is the coolant passage and the second influencing factor is the center line distance for positioning the coolant passage with respect to the profile. Shen S. et al. [38] studied thermal and mechanical performance using finite element analysis in a system with three CCCs with different profiles and the same aspect ratio. In addition, the results were validated experimentally. Compared with conventional straight channels, CCCs reduced cooling time ranging from 10% to 57% and provided a more uniform temperature distribution [38]. Regarding the economic analyses reported in the literature [39,40,41], CCC implementation results in improved performance of products. Various cost factors are considered for cost estimation of plastic injection molding components, including the material, mold, processing and post-processing costs. The high cost of tooling is a major cost component and differs between the use of straight drilled conventional cooling channels and the use of channels that conform to the shape of the part. For parts with complex shapes and high precision, it is always recommended to use conformal cooling channels as they reduce defects and cycle time. Singh et al. [40] studied the cost models for plastic injection molded components using conformal cooling channels versus those using conventional cooling channels. This study used a spray bottle funnel and the results showed a reduction of approximately 12% when using CCCs. 

The primary objective of this work is to review and analyze the effects of conformal and traditional design cooling channels on part quality and process efficiency. A design of experiments approach was implemented and all conditions were simulated using injection molding software. This study also explores the impact of mold materials with distinctly different thermal conductivities, such as metal and polymer (different stiffness conditions), in various CCC configurations compared with conventional cooling channels and easily deformable geometries. Based on the literature review, the response variables related to the injection molding process and the injected part are shown in Figure 1c, and the variables chosen for this study are marked with asterisks. This analysis provides the criteria needed to optimize mold cooling, a critical step in reducing shrinkage and warpage in molded parts. Such optimization significantly improves overall part quality by reducing sink defects and internal stresses.

## 2. Materials and Methods

The current study was designed following the scheme depicted in Figure 2. The CCCs were compared against conventional cooling channels. To achieve this, the initial analysis focused on selecting a specific CCC configuration from various options. Multiple simulations of the injection molding process were carried out to mold a cup-shaped part, using two distinct mold materials under specific processing conditions. The response variables—cooling efficiency, mold temperature distribution, and warpage—were considered to evaluate the performance of each mold configuration. Subsequently, the chosen CCC and the conventional cooling channels underwent simulations to compare the performance of each mold based on the aforementioned response variables. For this final analysis, an additional geometry with lower rigidity was evaluated to discern the impact on the quality of the plastic parts.

### 2.1. Conventional Cooling Channels

For the conventional system, a circular cross-section was used in the mold cavity and a baffle type was used in the mold core, as shown in Figure 3. The dimensions of the cooling channels, diameter, step and distance to the injected part were selected based on the results of the experimental design (Section 2.2).

### 2.2. Experimental Design of Conformal Cooling Channels

To determine the optimal CCC configuration, an experimental design approach was applied. It involved three geometric variables, aiming to assess their performance in terms of cooling efficiency, maximum warpage, and mold temperature distribution. The geometry chosen for this study was a cup-shaped part with upper and lower diameters of 54 and 48 mm, respectively, a height of 23 mm and a thickness of 3 mm. This part was selected because it is a simple geometric element that requires high volume production and homogeneous cooling to avoid significant geometric distortion. Digital models of the mold core, cavity and CCCs were created using a CAD software.

Two different geometries of the CCC cross-section were considered for the simulations: circular and square. Three variables with three levels were chosen for the DOE analysis to simulate the injection molding process using Moldex3D. Therefore, the DOE of the injection molding process was performed using a factorial design 3^3^. Minitab statistical software was used for the DOE analysis. A schematic representation of the variables is shown in Figure 4. D is the diameter—or side (S) in the case of a square channel—of the CCC, L is the distance between the center of the CCC and the outer wall of the molded part, and P (step) is the vertical distance between the CCCs. The levels for each variable were chosen based on the literature review and some initial simulations, as shown in Table 1.

### 2.3. Simulations

The geometry of the part was discretized into a mesh for the simulations using finite 3D tetrahedral elements. The number of elements in the mesh through the thickness of the part ranged from 6 to 8. The purpose of this level of refinement is to produce more accurate results. This number of mesh elements through the thickness is sufficient to achieve satisfactory simulation results, according to similar studies [42,43,44]. The simulations were performed using Moldex 3D^®^, for the mold in both materials (steel and polycarbonate). The thermal and density properties are shown in Table 2. In regard to the material properties of the steel mold (isotropic material, linear elastic), it has a Young’s modulus of 63,000 MPa and a Poisson coefficient of 0.33 [45,46]. For the polycarbonate, Young’s modulus is 2300 MPa [46]. Table 3 summarizes the processing conditions for each mold based on previous simulations and recommended injection molding processing conditions for injecting polypropylene (PP). The cooling time required to reach the ejection temperature of PP at recommended processing temperatures was estimated using the analytical Equation (1) [47,48].
polymers-15-04044-t002_Table 2Table 2Thermal properties and density for material molds and coolant [45,46,49].PropertyUnitsPolycarbonateWater (Coolant)SteelDensitykg/mm^3^1.20 × 10^−6^1.00 × 10^−6^7.56 × 10^−6^Specific HeatJ/(kg·K)12001000700ConductivityW/(mm·K)0.000220.0070.03
polymers-15-04044-t003_Table 3Table 3Injection process conditions.VariableUnitsPolycarbonateSteelMelt temperature°C220220Coolant temperature water°C3030Filling times22Packing times155Packing pressurebar90% of injection pressure90% of injection pressureCooling times7520
(1)$$Cooling\;time=\left(\frac{t^{2}}{\pi^{2}\times t_{dif}}\right)\times Ln\left(\frac{8\times \left(T_{m}-T_{c}\right)}{\pi^{2}\times \left(T_{e}-T_{c}\right)}\right)$$
where the notation is as follows:

*t*: injected part thickness.

*t_dif_*: Effective thermal diffusivity of the PP.

*T_m_*: Average melting temperature (°C).

*T_c_*: Average mold temperature (°C).

*T_e_*: Average ejection temperature of the injected part (°C).

## 3. Results

### 3.1. Simulation

Moldex 3D was used to simulate the injection process using the DOE for both geometries (circular and square). The results of the simulation for the steel and polycarbonate molds are shown in Table 4. While many parameters can be obtained from the simulation, two response variables were selected: the cooling efficiency difference between the dies, and the maximum warpage during the process. The cooling efficiency difference is calculated as the arithmetic difference between the cooling efficiency of the mold core cooling system and the mold cavity cooling system. As illustrated in Figure 5, the difference in cooling efficiency is 9.15%, which was calculated by subtracting 44.95% from 54.10%. Similarly, the cooling efficiency difference is computed for each simulation, as shown in Table 4.

Regarding the steel mold, it is possible to observe a high efficiency of heat dissipation via the CCC for the mold cavity and a lower efficiency (about 10% less) for the mold core. The maximum warpage of the injected part after simulation was about 0.6 mm. On the other hand, in the case of the polycarbonate mold, the difference in cooling efficiency and the maximum warpage were higher compared with the steel mold, about 30% and 2 mm, respectively. It is worth noting that the cooling efficiency was defined as the difference between the heat dissipation efficiency of the core and cavity of mold. Therefore, better and more homogeneous heat dissipation is achieved by reducing the difference in cooling efficiency.

Figure 6 shows the results for temperature distribution obtained from the simulations of each of the CCC configurations using steel and polycarbonate as molding materials. A higher mold temperature is often found toward the base of the mold core due to reduced efficiency of the cooling systems in this part of the mold, a pattern that is repeated in all the simulations. Additionally, mold cavity cooling systems are more efficient than mold core cooling systems (see Figure 5 and Table 4). This difference is reflected in the mold temperatures, which are higher in the core area and lower in the cavity. Upon examining the mold temperature distribution achieved by each CCC configuration, significantly lower temperature levels are evident within the metal mold compared with the polymer mold. This difference arises from the superior thermal conductivity of the metal, in contrast to the polymer (refer to Table 2). The simulation of the mold in polycarbonate, due to its low conductivity, required cooling times in the mold of more than 90 s (including injection and cooling time). However, the temperature levels in the mold walls rise to within a range of 80 to 100 °C. In regions where the cooling channels are more distant from the cavity and core surfaces, peak mold temperatures exceeding 180 °C are generated. Such conditions are not advisable for preserving the structural integrity of the polymer mold or for the thermal process of polypropylene injection molding. In a comparative study, Zink and Kovács assessed the injection of polypropylene into a polymer mold (Digital ABS Plus) with conductivity levels comparable to the polycarbonate utilized in this study. Despite implementing CCCs, the time required to regulate the mold temperature to the recommended level to inject polypropylene (40 °C) exceeded 200 s [46]. In contrast, the metal mold maintains reasonable temperatures between 30 and 45 °C for the mold core and between 46 and 60 °C for the mold cavity. The cooling time for the injected part in the metal mold was 20 s. Similar results were observed with square-section CCC configurations, as shown in Figure 7. In some cases, the square-section CCC configuration showed a slightly lower mold temperature than its circular-segment CCC counterpart. However, warpage levels do not vary significantly when comparing equivalent CCC configurations (Table 4).

Based on the results obtained in Figure 6 and Figure 7, the CCC configuration D6-P8-L6 generates a more homogeneous mold temperature distribution. This applies to both metal and polymer molds and for both circular and square channel sections (Figure 6e and Figure 7e). The mold temperatures achieved in the metal mold are in closest to the recommended mold temperatures for polypropylene injection, which range from 30 to 44.2 °C for circular-section D6-P8-L6 and 30 to 43.1 °C for square-section S6-P8-L6. This outcome is consistent with the findings presented in Table 4, where the D6-P8-L6 CCC configuration exhibits the smallest warpages and the least differences in core and cavity cooling efficiencies, observed for both circular and square channel sections. This relationship between cooling efficiency and final warpage has been corroborated by the research conducted by Kuo and Xu. They found that increasing cooling efficiency in the cores (resulting in the smallest difference in cooling efficiency between cores and cavities) resulted in a reduction in warpage of the injected plastic part [47].

Another important aspect to emphasize in the results is the limited sensitivity of the metal mold to changes in the CCC configuration in terms of its effect on the final warpage. The ultimate warpage level consistently falls within the range of 0.61 to 0.66 mm, with a 7.6% variation (see Table 4). This resilience is maintained despite possible discrepancies in cooling efficiency between core and cavity molds, or fluctuations in the final mold temperature in the different CCCs evaluated. However, in the polymer mold, the maximum and minimum warpages obtained were 2.05 and 1.47 mm, respectively, equivalent to a variation of 28.29% (refer to Table 4). This discrepancy underscores the significant impact of cooling efficiency differences between core and cavity molds (Table 4). Similarly, based on the CCC configuration employed, the end mold temperatures exhibit considerable variation. Notably, the CCC configuration D4-P16-L12 recorded peak values of 214 °C, and the S4-P16-L12 configuration reached a maximum of 210 °C (Figure 6d and Figure 7d). In contrast, the D6-P8-L6 and S6-P8-L6 configurations maintained a lower temperature of 171 °C (Figure 6e and Figure 7e). In essence, the choice of CCC configuration can have a significant impact on the quality of the injected part, as evidenced by the final warpage of the part when injected into the polymer mold.

### 3.2. Experimental Design (ANOVA Analysis)

As mentioned above, the simulation results were used as input for the DOE analysis. For each response variable, an ANOVA analysis was performed based on a 95% confidence interval. Therefore, a *p*-value less than 0.05 and a higher F-value indicate a parameter or condition with significance in the response variable. Table 5 displays an example of the analysis of variance for the maximum displacement after the injection process. It can be observed that variable L has a significant importance. In addition, the other variables and the two-way interaction are not relevant. In general, the variables L and D showed a significant effect in the response variables for both systems (steel and polycarbonate molds). In addition, an optimization response was generated for each variable, using a minimum target for both variables (warpage displacement and cooling efficiency difference), and the results showed that the configuration D6, P8 and L6 is **most optimal.**

Using the simulation results obtained and conducting an ANOVA analysis, we proceeded to perform complementary simulations for the D6-P8-L6 configuration. While additive manufacturing allows for the production of square sections without difficulties, circular sections were preferred. This decision was driven by the comparable differences in cooling efficiency and final warpage observed with square CCCs. In addition, the likelihood of mechanical failure in the acute areas of the square channels within the polymer molds is higher than in the case of circular sections. This increased mechanical reliability influenced the choice of circular section channels for the CCC D6-P8-L6.

In the following analysis, the performance of the CCC D6-P8-L6 was compared with a conventional cooling system. Furthermore, the cooling effects of different geometries were evaluated. A geometry previously evaluated in the initial DOE analysis was compared with an alternative configuration characterized by less rigid geometric features in both mold materials (steel and polycarbonate). For these specific mold configurations, variables such as maximum mold temperature, final part warpage, and mold temperature distribution were examined. Figure 8 illustrates the results of the mold temperature distribution obtained in the simulations. It is evident that the CCCs provide a more uniform and tightly distributed mold temperature, as recommended for polypropylene injection. This is particularly noticeable in the case of the metal mold (Figure 8a,c) when compared with the conventional cooling system. In the conventional system, there is a significant temperature difference between the core and cavity molds, a pattern observed for both types of injected geometries (Figure 8b,d). This is due to greater heat retention in the mold core because its geometry makes it difficult for heat to dissipate in this region. Several research studies with similar geometries have demonstrated that implementing CCCs in the core mold significantly enhances cooling efficiency compared with conventional channels [47,48]. As shown in previous simulations, the mold temperatures for the polycarbonate mold are substantially higher than those for the metal mold, with an extra rise when employing a conventional cooling system.

It is well known that uneven temperatures within the mold, or a significant temperature difference between the core and cavity components, will result in uneven cooling rates within the part. These variations manifest as differential material shrinkage, ultimately causing warpage in the part [50,51]. Based on the information provided by the mold temperature distribution and the final warpage report of the simulated part, Figure 9 and Figure 10 compare the values of these variables for each cooling system, type of injected part, and mold material.

In the case of the cup-shaped part, a more significant effect of CCCs is observed on the maximum temperature of the mold when using the metal mold. This results in a reduction of 64.1 °C to 44.5 °C in the maximum temperature when utilizing CCCs, corresponding to an 8.96% reduction in warpage compared with the mold with conventional cooling (Figure 9). Different behavior is observed in the polycarbonate mold. There is no significant difference in the maximum mold temperature when using either the conventional cooling system or CCCs, with a variation of only 0.81 °C. However, the use of CCCs results in a 9.26% reduction in warpage. These results demonstrate that the injected geometry can withstand warpages of up to 1.62 mm, even under extremely high-temperature conditions and non-uniform mold temperature distributions. This phenomenon is particularly notable in the context of low thermal conductivity polycarbonate molds, and it can be attributed to the inherent rigidity of the injection-molded product. Accordingly, we simulated the effect of injecting a geometry with lower rigidity by evaluating two cooling systems: conventional and CCCs (Figure 8c,d). The results for the maximum mold temperature and final warpage of the injected part with the lower rigidity geometry are presented in Figure 10. Comparing these with the results in Figure 9, a similar trend is observed, with the difference that the maximum temperature levels of the mold are lower. This can be explained by the fact that the cooling in the core with this new geometry allows for a greater heat evacuation by conduction in the mold, substantially reducing the temperature peaks of the mold that were observed in the closed geometric part. Despite this, warpage levels are higher compared to those of the closed-cover geometry, reaching a maximum of 4.25 mm in the polycarbonate mold with conventional cooling. This result confirms the lower rigidity of the injected part geometries. In this case, it can be observed that the use of CCCs is the optimal approach for reducing the final warpages of this type of part. When utilizing CCCs in the polycarbonate mold, the warpage in the plastic component measures only 1.71 mm, which is significantly lower than that obtained with traditional cooling (4.25 mm). Regarding the metal mold, the use of CCCs reduces the variation in warpage in plastic parts to 0.69 mm, whereas conventional cooling results in 1.02 mm of warpage. Therefore, implementing CCCs leads to a 32.4% reduction in warpage.

## 4. Conclusions

The results of the present study are consistent with findings from other studies regarding the advantages of employing CCCs in comparison with conventional channel cooling systems in injection molds. This approach results in improved cooling efficiency and more uniform mold temperature distribution, even in cores where heat dissipation is challenging. In addition, this study examines the effectiveness of conformal cooling channels (CCCs) in polymer mold materials. These materials are commonly used in rapid tooling, but their inherent low thermal conductivity undermines cooling efficiency. In this context, it has been clearly demonstrated that the integration of CCCs is of paramount importance to achieve a more balanced mold cooling process. Consequently, this yields a noteworthy 9.26% reduction in final warpage in the cup-shaped part when compared with the outcomes obtained through conventional cooling methods. Even more significant is the profound effect that CCCs can have on plastic parts with lower geometric stiffness. In the latter case, the application of CCCs reduced final part warpage by 32.4% in metal molds and by 59.8% in polymer molds. Although the use of CCCs in polymer molds can significantly reduce part warpage, for part thicknesses greater than 2 mm, cooling times greater than 100 s may be explored to achieve better mold temperature control at levels recommended for injection molded plastics such as polypropylene. These results highlight the need for implementing conformal cooling channels in injection molds when minimal part distortion is required, or to enhance cooling efficiency in molds made of materials with low thermal conductivity, such as polymeric molds. This is often the case in rapid tooling applications.

## Figures and Tables

**Figure 1 polymers-15-04044-f001:**
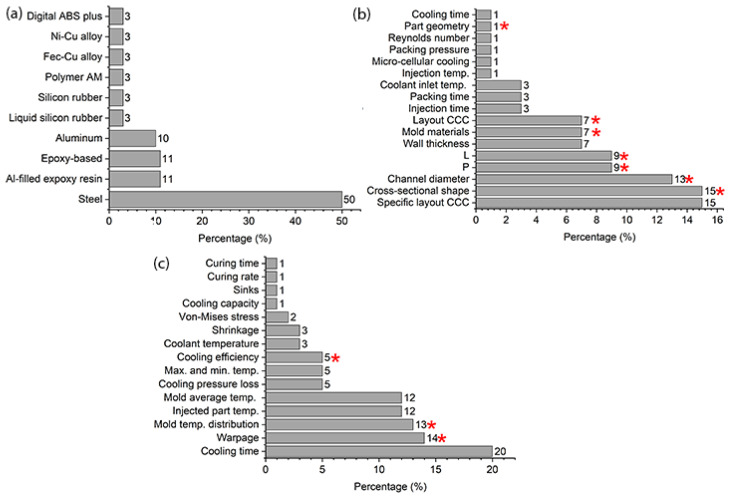
(**a**) CCC simulation and evaluation based on injection mold materials. (**b**) CCC design parameters (design parameters of this study marked with an asterisk). (**c**) Response variables in injection mold simulations (response variables used in this study marked with an asterisk).

**Figure 2 polymers-15-04044-f002:**
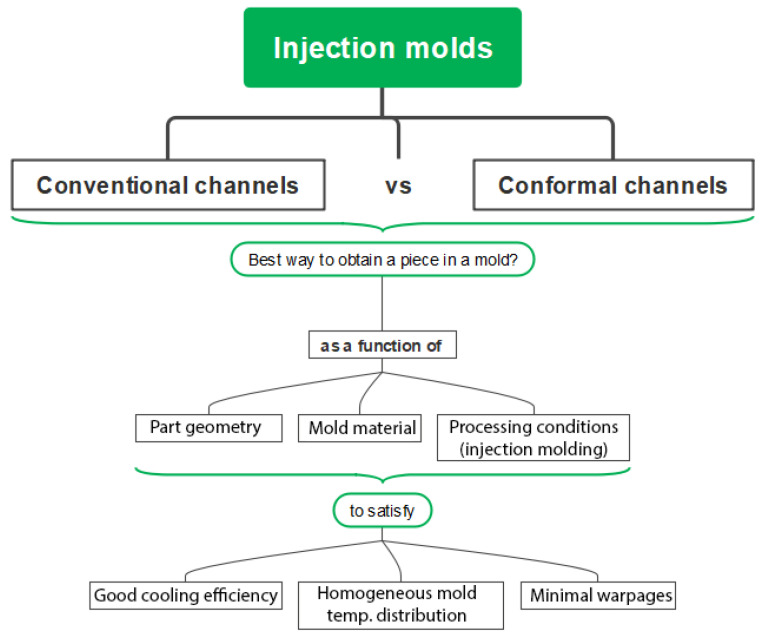
Analysis scheme for design and simulation of traditional vs. conformal channel geometries.

**Figure 3 polymers-15-04044-f003:**
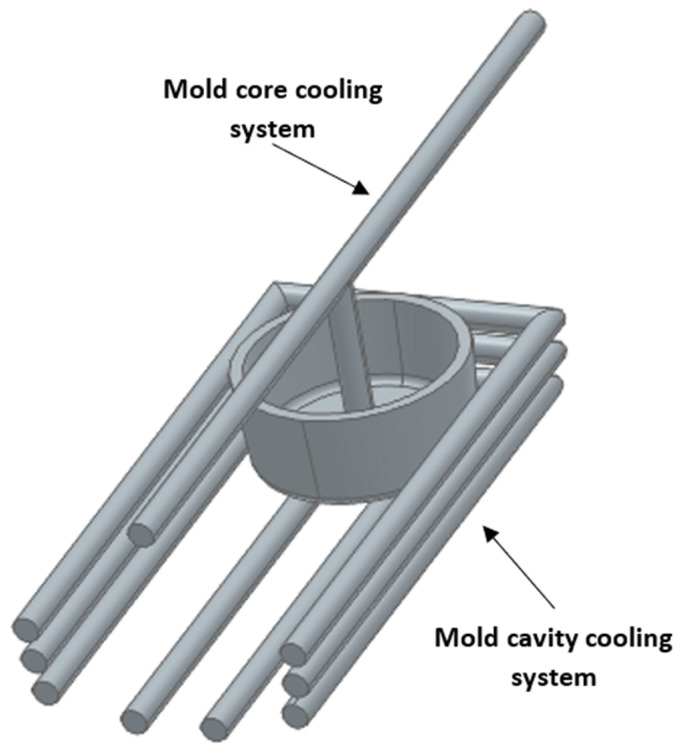
Configuration of conventional cooling channels for the core and cavity in the mold.

**Figure 4 polymers-15-04044-f004:**
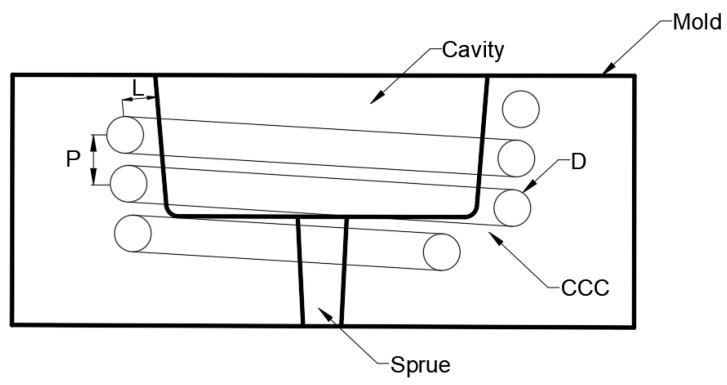
Schematic representation cross-section of the mold showing the variables for the DOE analysis. D—diameter, P—step, L—wall distance.

**Figure 5 polymers-15-04044-f005:**
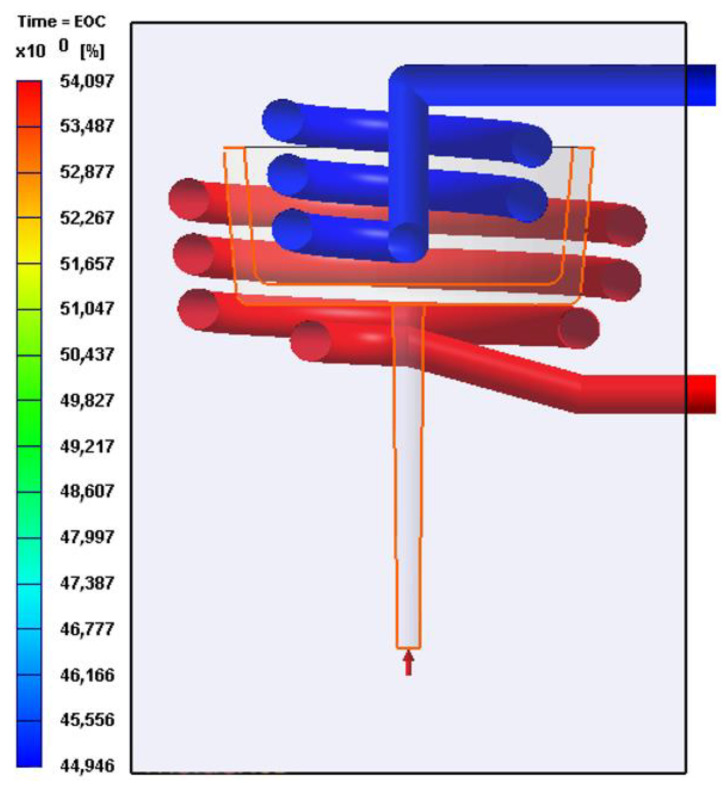
Cooling efficiency example for the circular CCC.

**Figure 6 polymers-15-04044-f006:**
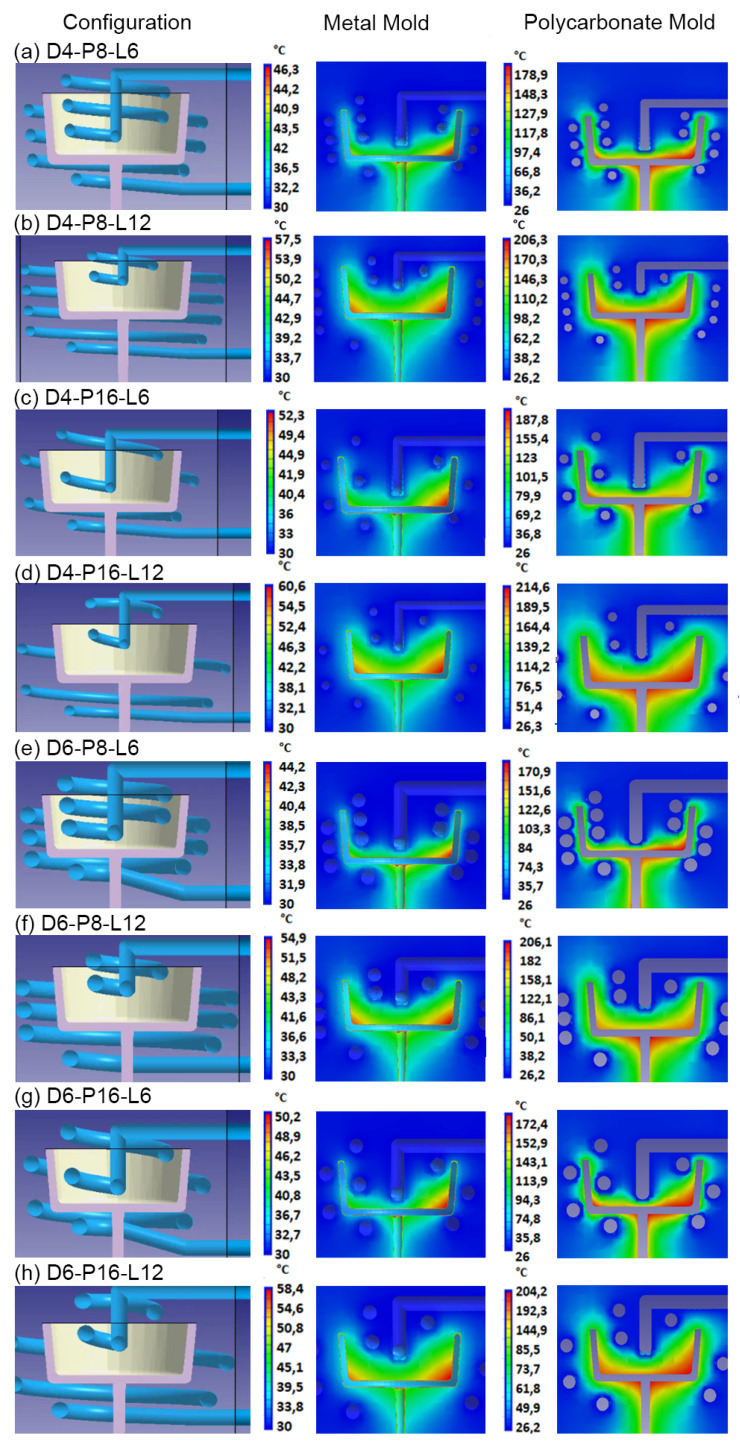
Circular CCC and mold temperature distribution for metallic mold and polycarbonate mold.

**Figure 7 polymers-15-04044-f007:**
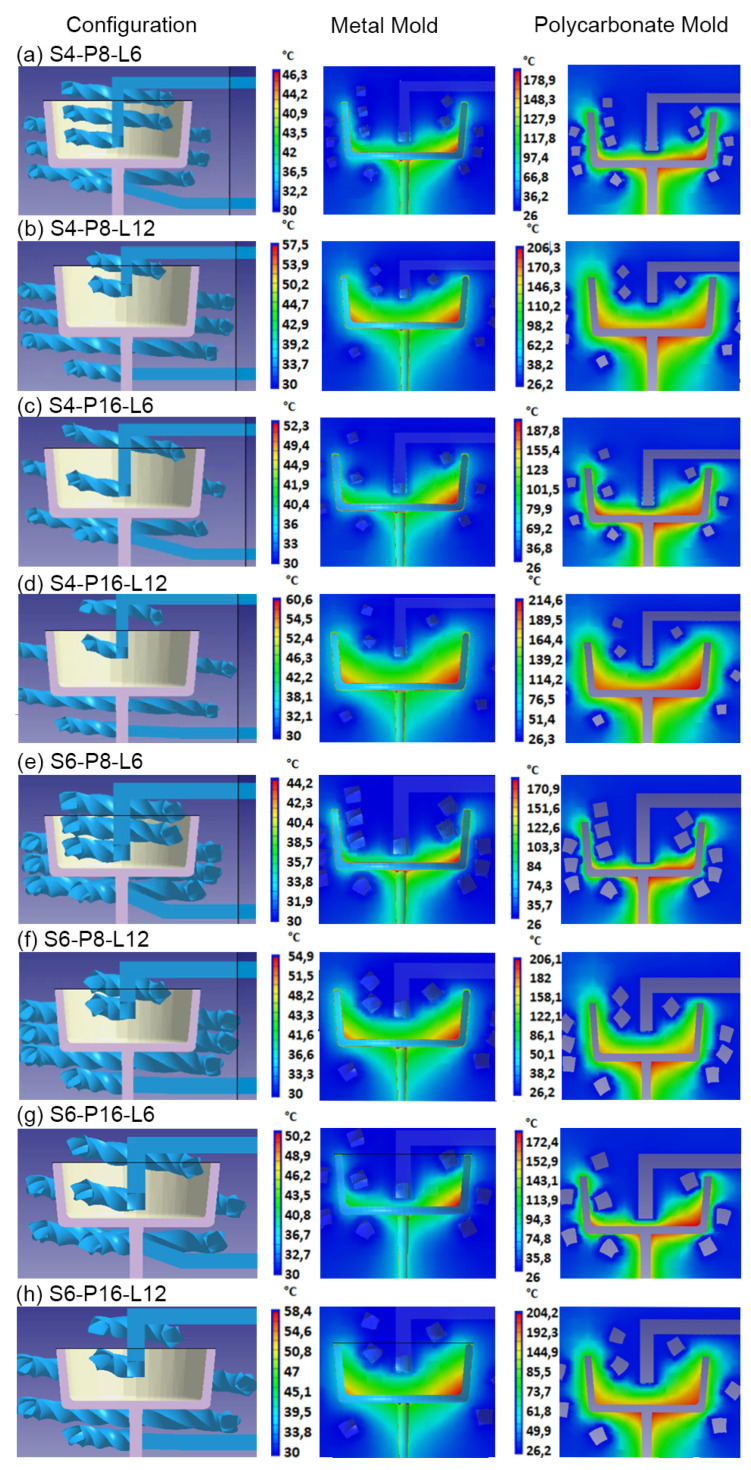
Rectangular CCC and mold temperature distribution for metallic mold and polycarbonate mold.

**Figure 8 polymers-15-04044-f008:**
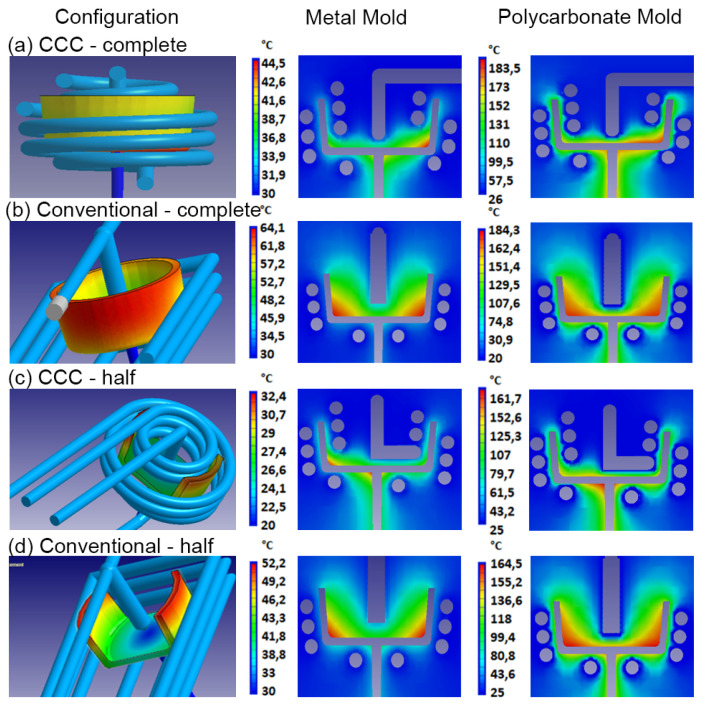
Evaluation of CCCs and conventional cooling channels for two different injected plastic parts as a function of mold temperature distribution for metallic mold and polycarbonate mold.

**Figure 9 polymers-15-04044-f009:**
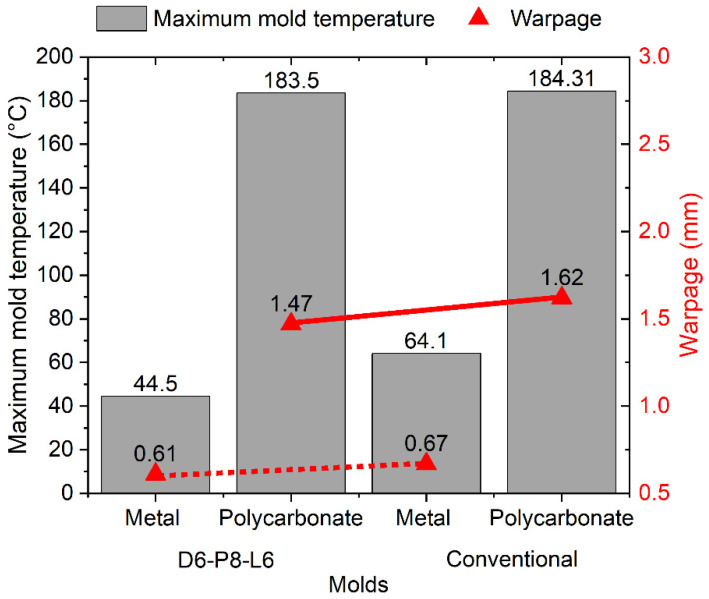
Maximum mold temperature and warpage for injected cup-shaped part.

**Figure 10 polymers-15-04044-f010:**
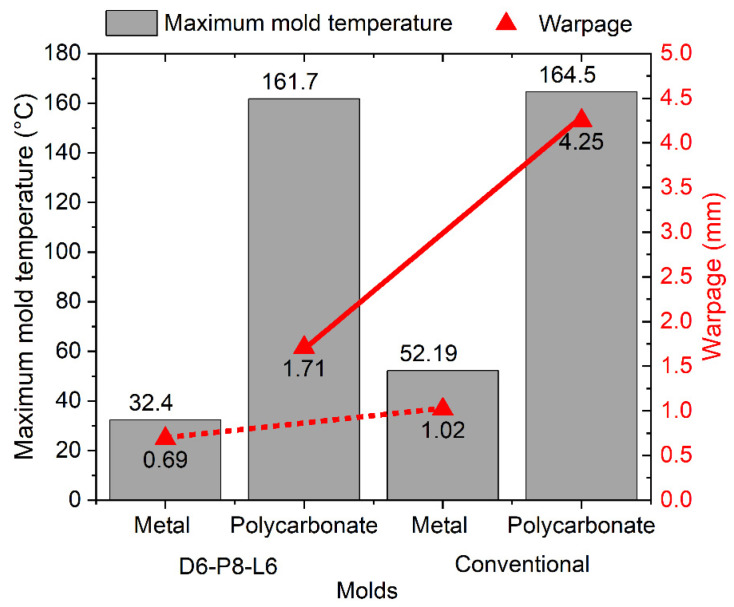
Maximum mold temperature and warpage for injected plastic part with lower rigidity geometry.

**Table 1 polymers-15-04044-t001:** Variables and levels for the DOE analysis.

Configuration	Diameter (D)–Size (S) (mm)	Step (P) (mm)	Wall Distance (L) (mm)	Experiment Number
Circular	4	8	6	1
			12	2
		16	6	3
			12	4
	6	8	6	5
			12	6
		16	6	7
			12	8
Square	4	8	6	9
			12	10
		16	6	11
			12	12
	6	8	6	13
			12	14
		16	6	15
			12	16

**Table 4 polymers-15-04044-t004:** Simulation results obtained from the Moldex 3D software.

Configuration	Test	Steel Mold		Polycarbonate Mold	
		Cooling Efficiency Difference (%)	Warpage Max. (mm)	Cooling Efficiency Difference (%)	Warpage Max. (mm)
Circular	D4-P8-L6	10.15	0.616	28.02	1.56
	D4-P8-L12	18.92	0.644	36.66	1.94
	D4-P16-L6	10.43	0.625	34.27	1.81
	D4-P16-L12	15.91	0.655	33.86	1.95
	D6-P8-L6	10.05	0.611	23.51	1.47
	D6-P8-L12	15.92	0.637	35.15	2.001
	D6-P16-L6	9.59	0.611	30.12	1.60
	D6-P16-L12	15.96	0.647	36.97	1.90
Square	S4-P8-L6	8.43	0.616	20.71	1.58
	S4-P8-L12	16.22	0.643	34.93	1.96
	S4-P16-L6	8.74	0.617	27.99	1.79
	S4-P16-L12	17.94	0.652	38.12	1.91
	S6-P8-L6	7.06	0.611	14.20	1.48
	S6-P8-L12	14.07	0.636	34.57	2.05
	S6-P16-L6	7.60	0.618	21.69	1.58
	S6-P16-L12	15.18	0.643	38.56	1.95

**Table 5 polymers-15-04044-t005:** Analysis of variance for warpage displacement using a steel mold and a circular geometry for the CCCs.

Source	DF	Adj SS	Adj MS	F-Value	*p*-Value
Model	6	0.002090	0.000348	43.53	0.116
Linear	3	0.002057	0.000686	85.71	0.079
L	1	0.001800	0.001800	225.00	0.042
P	1	0.000113	0.000113	14.06	0.166
D	1	0.000144	0.000144	18.06	0.147
2-Way Interactions	3	0.000033	0.000011	1.35	0.547
LxP	1	0.000018	0.000018	2.25	0.374
LxD	1	0.000002	0.000002	0.25	0.705
PxD	1	0.000013	0.000013	1.56	0.430
Error	1	0.000008	0.000008		
Total	7	0.002098			

## Data Availability

The data presented in this study are available on request from the corresponding author.

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
