# Peer review of "Evaluating the Cooling Efficiency of Polymer Injection Molds by Computer Simulation Using Conformal Channels"

_polymers, 2023, doi:10.3390/polym15204044_

Round 1

Reviewer 1 Report

This study investigated the conformal cooling channel effect on injection mold production with computational simulation. Comparative studies and statistical analysis have been performed to demonstrate the improvement from CCC design. This manuscript is well written and structured. The results and findings are well presented and easy to follow. This study can be helpful for optimizing the efficiency of plastic production by refining the cooling channel designs.

(1) Relevant references are recommended to support statements in the introduction. For example, line #49, “computerized simulation has proven…”, and line #129, “…CCC reduce cooling time by 10% to 57% and provide more uniform temperature distribution…”.

(2) When selecting the variables to be considered in the study, line#78 “In contrast to previous studies… the present study evaluates the parameters shown in Figure 2 (marked with asterisk)”. Could the authors provide some justification for the selection of these parameters?

(3) The Introduction section is a bit lengthy; it is recommended to condense the content.

(4) A definition of “Cooling efficiency difference” needs to be provided.

(5) In the Conclusion section, line #362 states that “the application of CC reduced final part warpage by 32.4% in metal molds and by 59.7% in polymer molds”. It is recommended to bring up the analysis of these numbers in the main body part before the final conclusion.

Reviewer 2 Report

This work used computer simulation to demonstrate the improvement of cooling efficiency using conformal cooling channels compared with conventional cooling channels. I think overall, the authors presented their research design clearly with sufficient evidence to support their conclusions. I only have minor suggestions.

1. I think Figure 1-3 can be consolidated into one figure with multiple subplots which should make the paper more concise.

2. I was wondering if the authors could change the font in Figure 4, especially the lower part which is hard to read in current font.

3. In Table 2, the equation for Cooling time, I was wondering if the authors could change the use of ‘x’ to maybe ‘ ‘, or not use the symbol at all. I assume ‘x’ is representing multiplication. Right now, it makes the equation hard to read.

4. I was wondering if the authors could use a higher resolution image for Figure 7.

5. In Figure 10c, for Metal Mold and Polycarbonate Mold, I was wondering why the images are not showing the two side walls for cavity like in Figure 10d.

6. In conclusion, I was wondering if the authors could make some comments on the implications or applications of their findings in addition to summarize all the results shown before.

7. In line 20, there is an extra whitespace after ‘48’. In line 252, there is an extra whitespace in front of ‘In some cases’.
